# Variational Learning on Aggregate Outputs with Gaussian Processes

**Ho Chung Leon Law**[*]
University of Oxford

**Dino Sejdinovic**[†]
University of Oxford

**Ewan Cameron**[‡]
University of Oxford

**Tim CD Lucas**[‡]
University of Oxford

**Seth Flaxman**[§]
Imperial College London

**Katherine Battle**[‡]
University Of Oxford

**Kenji Fukumizu**[¶]
Institute of Statistical Mathematics

## Abstract

While a typical supervised learning framework assumes that the inputs and the outputs are measured at the same levels of granularity, many applications, including global mapping of disease, only have access to outputs at a much coarser level than that of the inputs. Aggregation of outputs makes generalization to new inputs much more difficult. We consider an approach to this problem based on variational learning with a model of output aggregation and Gaussian processes, where aggregation leads to intractability of the standard evidence lower bounds. We propose new bounds and tractable approximations, leading to improved prediction accuracy and scalability to large datasets, while explicitly taking uncertainty into account. We develop a framework which extends to several types of likelihoods, including the Poisson model for aggregated count data. We apply our framework to a challenging and important problem, the fine-scale spatial modelling of malaria incidence, with over 1 million observations.

## 1   Introduction

A typical supervised learning setup assumes existence of a set of input-output examples $\{(x_\ell, y_\ell)\}_\ell$ from which a functional relationship or a conditional probabilistic model of outputs given inputs can be learned. A prototypical use-case is the situation where obtaining outputs $y_\star$ for new, previously unseen, inputs $x_\star$ is costly, i.e., labelling is expensive and requires human intervention, but measurements of inputs are cheap and automated. Similarly, in many applications, due to a much greater cost in acquiring labels, they are only available at a much coarser resolution than the level at which the inputs are available and at which we wish to make predictions. This is the problem of *weakly supervised* learning on aggregate outputs [14, 20], which has been studied in the literature in a variety of forms, with classification and regression notably being developed separately and without any unified treatment which can allow more flexible observation models. In this contribution, we consider a framework of observation models of aggregated outputs given bagged inputs, which reside in exponential families. While we develop a more general treatment, the main focus in the paper is on the Poisson likelihood for count data, which is motivated by the applications in spatial statistics. In particular, we consider the important problem of fine-scale mapping of diseases. High resolution maps of infectious disease risk can offer a powerful tool for developing National Strategic Plans,

---

[*]Department of Statistics, Oxford, UK. <ho.law@stats.ox.ac.uk>

[†]Department of Statistics, Oxford, UK. Alan Turing Institute, London, UK. <dino.sejdinovic@stats.ox.ac.uk>

[‡]Big Data Institute, Oxford, UK. <dr.ewan.cameron@gmail.com, timcdlucas@gmail.com, katherine.battle@bdi.ox.ac.uk>

[§]Department of Mathematics and Data Science Institute, London, UK. <s.flaxman@imperial.ac.uk>

[¶]Tachikawa, Tokyo, Japan. <fukumizu@ism.ac.jp>

allowing accurate stratification of intervention types to areas of greatest impact [5]. In low resource settings these maps must be constructed through probabilistic models linking the limited observational data to a suite of spatial covariates (often from remote-sensing images) describing social, economic, and environmental factors thought to influence exposure to the relevant infectious pathways. In this paper, we apply our method to the incidence of clinical malaria cases. Point incidence data of malaria is typically available at a high temporal frequency (weekly or monthly), but lacks spatial precision, being aggregated by administrative district or by health facility catchment. The challenge for risk modelling is to produce fine-scale predictions from these coarse incidence data, leveraging the remote-sensing covariates and appropriate regularity conditions to ensure a well-behaved problem.

Methodologically, the Poisson distribution is a popular choice for modelling count data. In the mapping setting, the intensity of the Poisson distribution is modelled as a function of spatial and other covariates. We use Gaussian processes (GPs) as a flexible model for the intensity. GPs are a widely used approach in spatial modelling but also one of the pillars of Bayesian machine learning, enabling predictive models which explicitly quantify their uncertainty. Recently, we have seen many advances in variational GP posterior approximations, allowing them to couple with more complex observation likelihoods (e.g. binary or Poisson data [21, 17]) as well as a number of effective scalable GP approaches [24, 30, 8, 9], extending the applicability of GPs to dataset sizes previously deemed prohibitive.

**Contribution** Our contributions can be summarised as follows. A general framework is developed for *aggregated observation models* using exponential families and Gaussian processes. This is novel, as previous work on aggregation or bag models focuses on specific types of output models such as binary classification. Tractable and scalable variational inference methods are proposed for several instances of the aggregated observation models, making use of novel lower bounds on the model evidence. In experiments, it is demonstrated that the proposed methods can scale to dataset sizes of more than 1 million observations. We thoroughly investigate an application of the developed methodology to disease mapping from coarse measurements, where the observation model is Poisson, giving encouraging results. Uncertainty quantification, which is explicit in our models, is essential for this application.

**Related Work** The framework of learning from aggregate data was believed to have been first introduced in [20], which considers the two regimes of classification and regression. However, while the task of classification of individuals from aggregate data (also known as *learning from label proportions*) has been explored widely in the literature [23, 22, 13, 18, 35, 34, 14], there has been little literature on the analogous regression regime in the machine learning community. Perhaps the closest literature available is [13], who considers a general framework for learning from aggregate data, but also only considers the classification case for experiments. In this work, we will appropriately adjust the framework in [13] and take this to be our baseline. A related problem arises in the spatial statistics community under the name of 'down-scaling', 'fine-scale modelling' or 'spatial disaggregation' [11, 10], in the analysis of disease mapping, agricultural data, and species distribution modelling, with a variety of proposed methodologies (cf. [33] and references therein), including kriging [6]. However, to the best of our knowledge, approaches making use of recent advances in scalable variational inference for GPs are not considered.

Another closely related topic is *multiple instance learning* (MIL), concerned with classification with max-aggregation over labels in a bag, i.e. a bag is positively labeled if at least one individual is positive, and it is otherwise negatively labelled. While the task in MIL is typically to predict labels of new unobserved *bags*, [7] demonstrates that individual labels of a GP classifier can also be inferred in MIL setting with variational inference. Our work parallels that approach, considering bag observation models in exponential families and deriving new approximation bounds for some common generalized linear models. In deriving these bounds, we have taken an approach similar to [17], who considers the problem of Gaussian process-modulated *Poisson process* estimation using variational inference. However, our problem is made more complicated by the aggregation of labels, as standard lower bounds to the marginal likelihood used in [17] are also intractable in our model. Other related research topics include distribution regression and set regression, as in [28, 15, 16] and [36]. In these regression problems, while the input data for learning is the same as the current setup, the goal is to learn a function at the bag level, rather than the individual level, the application of these methods in our setting, naively treating single individuals as "distributions", may lead to suboptimal

performance. An overview of some other approaches for classification using bags of instances is given in [4].

## 2 Bag observation model: aggregation in mean parameters

Suppose we have a statistical model $p(y|\eta)$ for output $y \in \mathcal{Y}$, with parameter $\eta$ given by a function of input $x \in \mathcal{X}$, i.e., $\eta = \eta(x)$. Although one can formulate $p(y|\eta)$ in an arbitrary fashion, practitioners often only focus on interpretable simple models, hence we restrict our attention to $p(y|\eta)$ arising from exponential families. We assume that $\eta$ is the mean parameter of the exponential family.

Assume that we have a fixed set of points $x_i^a \in \mathcal{X}$ such that $\mathbf{x}^a = \{x_1^a, \ldots, x_{N_a}^a\}$ is a *bag* of points with $N_a$ *individuals*, and we wish to estimate the regression value $\eta(x_i^a)$ for each individual. However, instead of the typical setup where we have a paired sample $\{(x_\ell, y_\ell)\}_\ell$ of individuals and their outputs to use as a training set, we observe only *aggregate outputs* $y^a$ for each of the bags. Hence, our training data is of the form

$$(\{x_i^1\}_{i=1}^{N_1}, y^1), \ldots (\{x_i^n\}_{i=1}^{N_n}, y^n), \tag{1}$$

and the goal is to estimate parameters $\eta(x_i^a)$ corresponding to individuals. To relate the aggregate $y^a$ and the bag $\mathbf{x}^a = (x_i^a)_{i=1}^{N_a}$, we use the following *bag observation model*:

$$y^a | \mathbf{x}^a \sim p(y|\eta^a), \qquad \eta^a = \sum_{i=1}^{N_a} w_i^a \eta(x_i^a), \tag{2}$$

where $w_i^a$ is an optional fixed non-negative weight used to adjust the scales (see Section 3 for an example). Note that the aggregation in the bag observation model is on the mean parameters for individuals, not necessarily on the individual responses $y_i^a$. This implies that each individual contributes to the mean bag response and that the observation model for bags belongs to the same parametric form as the one for individuals. For tractable and scalable estimation, we will use variational methods, as the aggregated observation model leads to intractable posteriors. We consider the Poisson, normal, and exponential distributions, but devote a special focus to the Poisson model in this paper, and refer readers to Appendix A for other cases and experimental results for the Normal model in Appendix H.2.

It is also worth noting that we place no restrictions on the collection of the individuals, with the bagging process possibly dependent on covariates $x_i^a$ or any unseen factors. The bags can also be of different sizes. After we obtain our individual model $\eta(x)$, we can use it for prediction of in-bag individuals, as well as out-of-bag individuals.

## 3 Poisson bag model: Modelling aggregate counts

The Poisson distribution $p(y|\lambda) = \lambda^y e^{-\lambda}/(y!)$ is considered for count observations, and this paper discusses the Poisson regression with intensity $\lambda(x_i^a)$ multiplied by a 'population' $p_i^a$, which is a constant assumed to be known for each individual (or 'sub-bag') in the bag. The population for a bag $a$ is given by $p^a = \sum_i p_i^a$. An observed bag count $y^a$ is assumed to follow

$$y^a | \mathbf{x}^a \sim \text{Poisson}(p^a \lambda^a), \quad \lambda^a := \sum_{i=1}^{N_a} \frac{p_i^a}{p^a} \lambda(x_i^a).$$

Note that, by introducing unobserved counts $y_i^a \sim \text{Poisson}(y_i^a | p_i^a \lambda(x_i^a))$, the bag observation $y^a$ has the same distribution as $\sum_{i=1}^{N_a} y_i^a$ since the Poisson distribution is closed under convolutions. If a bag and its individuals correspond to an area and its partition in geostatistical applications, as in the malaria example in Section 4.2, the population in the above bag model can be regarded as the population of an area or a sub-area. With this formulation, the goal is to estimate the basic intensity function $\lambda(x)$ from the aggregated observations (1). Assuming independence given $\{\mathbf{x}^a\}_a$, the negative log-likelihood (NLL) $\ell_0$ across bags is

$$-\log[\Pi_{a=1}^n p(y^a|\mathbf{x}^a)] \stackrel{c}{=} \sum_{a=1}^n p^a \lambda^a - y^a \log(p^a \lambda^a) \stackrel{c}{=} \sum_{a=1}^n \left[\sum_{i=1}^{N_a} p_i^a \lambda(x_i^a) - y^a \log\left(\sum_{i=1}^{N_a} p_i^a \lambda(x_i^a)\right)\right], \tag{3}$$

where $\stackrel{c}{=}$ denotes an equality up to additive constant. During training, this term will pass information from the bag level observations $\{y^a\}$ to the individual basic intensity $\lambda(x_i^a)$. It is noted that once we

have trained an appropriate model for $\lambda(x_i^a)$, we will be able to make individual level predictions, and also bag level predictions if desired. We will consider baselines with (3) using penalized likelihoods inspired by manifold regularization in semi-supervised learning [2] – presented in Appendix B. In the next section, we propose a model for $\lambda$ based on GPs.

### 3.1 VBAgg-Poisson: Gaussian processes for aggregate counts

Suppose now we model $f$ as a Gaussian process (GP), then we have:

$$y^a|\mathbf{x}^a \sim \text{Poisson}\left(\sum_{i=1}^{N_a} p_i^a \lambda_i^a\right), \qquad \lambda_i^a = \Psi(f(x_i^a)), \qquad f \sim GP(\mu, k) \tag{4}$$

where $\mu$ and $k$ are some appropriate mean function and covariance kernel $k(x, y)$. (For implementation, we consider a constant mean function.) Since the intensity is always non-negative, in all models, we will need to use a transformation $\lambda(x) = \Psi(f(x))$, where $\Psi$ is a non-negative valued function. We will consider cases $\Psi(f) = f^2$ and $\Psi(f) = e^f$. A discussion of various choices of this link function in the context of Poisson intensities modulated by GPs is given in [17]. Modelling $f$ with a GP allows us to propagate uncertainty on the predictions to $\lambda_i^a$, which is especially important in this weakly supervised problem setting, where we do not directly observe any individual output $y_i^a$. Since the total number of individuals in our target application of disease mapping is typically in the millions (see Section 4.2), we will approximate the posterior over $\lambda_i^a := \lambda(x_i^a)$ using variational inference, with details found in Appendix E.

For scalability of the GP method, as in previous literature [7, 17], we use a set of inducing points $\{u_\ell\}_{\ell=1}^m$, which are given by the function evaluations of the Gaussian process $f$ at landmark points $W = \{w_1, \ldots, w_m\}$; i.e., $u_\ell = f(w_\ell)$. The distribution $p(u|W)$ is thus given by

$$u \sim N(\mu_W, K_{WW}), \qquad \mu_W = (\mu(w_\ell))_\ell, \qquad K_{WW} = (k(w_s, w_t))_{s,t}. \tag{5}$$

The joint likelihood is given by:

$$p(y, f, u|X, W, \Theta) = \prod_{a=1}^n \prod_{i=1}^{N_a} \text{Poisson}(y^a|p^a\lambda^a)p(f|u)p(u|W), \text{ with } f|u \sim GP(\tilde{\mu}_u, \tilde{K}), \tag{6}$$

$$\tilde{\mu}(z) = \mu_z + \mathbf{k}_{zW} K_{WW}^{-1}(u - \mu_W), \qquad \tilde{K}(z, z') = k(z, z') - \mathbf{k}_{zW} K_{WW}^{-1} \mathbf{k}_{Wz'} \tag{7}$$

where here $\lambda^a$, $f$ depends on $i$ implicitly, $\mathbf{k}_{zW} = (k(z, w_1), \ldots, k(z, w_\ell))^T$, with $\mu_W$, $\mu_z$ denoting their respective evaluations of the mean function $\mu$ and $\Theta$ are parameters of the mean and kernel functions of the GP. Proceeding similarly to [17], which discusses (non-bag) Poisson regression with GP, we obtain a lower bound of the marginal log-likelihood $\log p(y|\Theta)$, introducing a variational distribution $q(u)$ (that we optimise):

$$\log p(y|\Theta) = \log \int \int p(y, f, u|X, W, \Theta)df\,du$$

$$\geq \int \int \log\left\{p(y|f, \Theta)\frac{p(u|W)}{q(u)}\right\}p(f|u, \Theta)q(u)df\,du \quad \text{(Jensen's inequality)}$$

$$= \sum_a \int \int \left\{y^a \log\left(\sum_{i=1}^{N_a} p_i^a \Psi(f(x_i^a))\right) - \left(\sum_{i=1}^{N_a} p_i^a \Psi(f(x_i^a))\right)\right\}p(f|u)q(u)df\,du$$

$$- \sum_a \log(y^a!) - KL(q(u)||p(u|W)) =: \mathcal{L}(q, \Theta), \tag{8}$$

The general solution to the maximization over $q$ of the evidence lower bound $\mathcal{L}(q, \Theta)$ above is given by the posterior of the inducing points $p(u|y)$, which is intractable. We introduce a restriction to the class of $q(u)$ to approximate the posterior $p(u|y)$. Suppose that the variational distribution $q(u)$ is Gaussian, $q(u) = N(\eta_u, \Sigma_u)$. We then need to maximize the lower bound $\mathcal{L}(q, \Theta)$ over the variational parameters $\eta_u$ and $\Sigma_u$.

The resulting $q(u)$ gives an approximation to the posterior $p(u|y)$ which also leads to a Gaussian approximation $q(f) = \int p(f|u)q(u)du$ to the posterior $p(f|y)$, which we finally then transform

through $\Psi$ to obtain the desired approximate posterior on each $\lambda(x_a^i)$ (which is either log-normal or non-central $\chi^2$ depending on the form of $\Psi$). The approximate posterior on $\lambda$ will then allow us to make predictions for individuals while, crucially, taking into account the uncertainties in $f$ (note that even the posterior predictive mean of $\lambda$ will depend on the predictive variance in $f$ due to the nonlinearity $\Psi$). We also want to emphasis the use of inducing variables is essential for scalability in our model: we cannot directly obtain approximations to the posterior of $\lambda(x_i^a)$ for all individuals, since this is often large in our problem setting (Section 4.2).

As the $p(u|W)$ and $q(u)$ are both Gaussian, the last term (KL-divergence) of (8) can be computed explicitly with exact form found in Appendix E.3. To consider the first two terms, let $q^a(v^a)$ be the marginal normal distribution of $v^a = (f(x_1^a), \dots, f(x_{N_a}^a))$, where $f$ follows the variational posterior $q(f)$. The distribution of $v^a$ is then $N(m^a, S^a)$, using (7) :

$$m^a = \mu_{\mathbf{x}^a} + K_{\mathbf{x}^a W} K_{WW}^{-1}(\eta_u - \mu_W), \ S^a = K_{\mathbf{x}^a, \mathbf{x}^a} - K_{\mathbf{x}^a W}\left(K_{WW}^{-1} - K_{WW}^{-1}\Sigma_u K_{WW}^{-1}\right)K_{W\mathbf{x}^a} \tag{9}$$

In the first term of (8), each summand is of the form

$$y^a \int \log\Big(\sum_{i=1}^{N_a} p_i^a \Psi\left(v_i^a\right)\Big)q^a(v^a)dv^a - \sum_{i=1}^{N_a} p_i^a \int \Psi\left(v_i^a\right)q^a(v^a)dv^a, \tag{10}$$

in which the second term is tractable for both of $\Psi(f) = f^2$ and $\Psi(f) = e^f$. The integral of the first term, however with $q^a$ Gaussian is not tractable. To solve this, we take different approaches for $\Psi(f) = f^2$ and $\Psi(f) = e^f$; for the former, approximation by Taylor expansion is applied, while for the latter, further lower bound is taken.

First consider the case $\Psi(f) = f^2$, and rewrite the first term of (8) as:

$$y^a \mathbb{E} \log \|V^a\|^2 \quad \text{, where } V^a \sim N(\tilde{m}^a, \tilde{S}^a),$$

with $P^a = diag\left(p_1^a, \dots, p_{N_a}^a\right), \tilde{m}^a = P^{a1/2}m^a$ and $\tilde{S}^a = P^{a1/2}S^a P^{a1/2}$. By a Taylor series approximation for $\mathbb{E} \log \|V^a\|^2$ (similar to [29]) around $\mathbb{E} \|V^a\|^2 = \|\tilde{m}^a\|^2 + tr\tilde{S}^a$, we obtain

$$\int \log\Big(\sum_{i=1}^{N_a} p_i^a(v_i^a)^2\Big)q^a(v^a)dv^a$$

$$\approx \log\left(m^{a\top}P^a m^a + tr(S^a P^a)\right) - \frac{2m^{a\top}P^a S^a P^a m^a + tr\left((S^a P^a)^2\right)}{\left(m^{a\top}P^a m^a + tr(S^a P^a)\right)^2} =: \zeta^a. \tag{11}$$

with details are in Appendix E.4. An alternative approach which we use for the case $\Psi(f) = e^f$ is to take a further lower bound, which is applicable to a general class of $\Psi$ (we provide further details for the analogous approach for $\Psi(v) = v^2$ in Appendix E.2). We use the following Lemma (proof found in Appendix E.1):

**Lemma 1.** *Let* $v = [v_1, \dots, v_N]^\top$ *be a random vector with probability density* $q(v)$ *with marginal densities* $q_i(v)$, *and let* $w_i \geq 0$, $i = 1, \dots, N$. *Then, for any non-negative valued function* $\Psi(v)$,

$$\int \log(\sum_{i=1}^N w_i \Psi(v_i))q(v)dv \geq \log\Big(\sum_{i=1}^N w_i e^{\xi_i}\Big), \quad where \quad \xi_i := \int \log \Psi(v_i)q_i(v_i)dv_i.$$

Hence we obtain that

$$\int \log(\sum_{i=1}^{N_a} p_i^a e^{v_i^a})q^a(v^a)dv^a \geq \log\Big(\sum_{i=1}^{N_a} p_i^a e^{m_i^a}\Big), \tag{12}$$

Using the above two approximation schemes, our objective (up to constant terms) can be formulated as: 1) $\Psi(v) = v^2$

$$\mathcal{L}_1^s(\Theta, \eta_u, \Sigma_u, W) := \sum_{a=1}^n y^a \zeta^a - \sum_{a=1}^n \sum_{i=1}^{N_a} \{(m_i^a)^2 + S_{ii}^a/2\} - KL(q(u)\|p(u|W)), \tag{13}$$

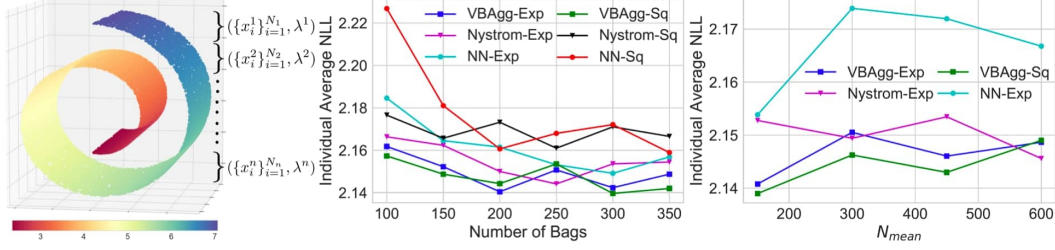

Figure 1: **Left**: Random samples on the Swiss roll manifold. **Middle, Right**: Individual Average NLL on train set for varying number of training bags $n$ and increasing $N_{mean}$, over 5 repetitions. Constant prediction within bag gives a NLL of 2.22. bag-pixel model gives NLL above 2.4 for the varying number of bags experiment.

2) $\Psi(v) = e^v$

$$\mathcal{L}_1^e(\Theta, \eta_u, \Sigma_u, W) := \sum_{a=1}^n y^a \log\Big(\sum_{i=1}^{N_a} e^{m_i^a}\Big) - \sum_{j=1}^n \sum_{i=1}^{N_a} e^{m_i^a + S_{ii}^a/2} - KL(q(u)\|p(u|W)). \quad (14)$$

Given these objectives, we can now optimise these lower bounds with respect to variational parameters $\{\eta_u, \Sigma_u\}$, parameters $\Theta$ of the mean and kernel functions, using stochastic gradient descent (SGD) on bags. Additionally, we might also learn $W$, locations for the landmark points. In this form, we can also see that the bound for $\Psi(v) = e^v$ has the added computational advantage of not requiring the full computation of the matrix $S^a$, but only its diagonals, while for $\Psi(v) = v^2$ computation of $\zeta^a$ involves full $S^a$, which may be problematic for extremely large bag sizes.

## 4 Experiments

We will now demonstrate various approaches: Variational Bayes with Gaussian Process (VBAgg), a MAP estimator of Bayesian Poisson regression with explicit feature maps (Nyström) and a neural network (NN) – the latter two employing manifold regularisation with RBF kernel (unless stated otherwise). For additional baselines, we consider a constant within bag model (constant), i.e. $\hat{\lambda}_i^a = \frac{y^a}{p^a}$ and also consider creating 'individual' covariates by aggregation of the covariates within a bag (bag-pixel). For details of all these approaches, see Appendix B. We also denote $\Psi(v) = e^v$ and $v^2$ as Exp and Sq respectively.

We implement our models in *TensorFlow*[6] and use SGD with Adam [12] to optimise their respective objectives, and we split the dataset into 4 parts, namely train, early-stop, validation and test set. Here the early-stop set is used for early stopping for the Nyström, NN and bag-pixel models, while the VBAgg approach ignores this partition as it optimises the lower bound to the marginal likelihood. The validation set is used for parameter tuning of any regularisation scaling, as well as learning rate, layer size and multiple initialisations. Throughout, VBAgg and Nyström have access to the same set of landmarks for fair comparison. It is also important to highlight that we perform early stopping and tuning based on *bag* level performance on NLL only, as this is the only information available to us.

For the VBAgg model, there are two approaches to tuning, one approach is to choose parameters based on NLL on the validation bag sets, another approach is to select all parameters based on the training objective $\mathcal{L}_1$, the lower bound to the marginal likelihood. We denote the latter approach VBAgg-Obj and report its toy experimental results in Appendix H.1.1 for presentation purposes. In general, the results are relatively *insensitive* to this choice, especially when $\Psi(v) = v^2$. To make predictions, we use the mean of our approximated posterior (provided by a log-normal and non-central $\chi^2$ distribution for Exp and Sq). As an additional evaluation, we report mean square error (MSE) and bag performance results in Appendix H.

## 4.1 Poisson Model: Swiss Roll

We first demonstrate our method on the swiss roll dataset[7], illustrated in Figure 1 (left). To make this an aggregate learning problem, we first construct $n$ bags with sizes drawn from a negative binomial distribution $N_a \sim NB(N_{mean}, N_{std})$, where $N_{mean}$ and $N_{std}$ represents the respective mean and standard deviation of $N_a$. We then randomly select $\sum_{a=1}^{n} N_a$ points from the swiss roll manifold to be the locations, giving us a set of colored locations in $\mathbb{R}^3$. Ordering these random locations by their $z$-axis coordinate, we group them, filling up each bag in turn as we move along the $z$-axis. The aim of this is to simulate that in real life the partitioning of locations into bags is often not independent of covariates. Taking the colour of each location as the underlying rate $\lambda_i^a$ at that location, we simulate $y_i^a \sim Poisson(\lambda_i^a)$, and take our observed outputs to be $y^a = \sum_{i=1}^{N_a} y_i^a \sim Poisson(\lambda^a)$, where $\lambda^a = \sum_{i=1}^{N_a} \lambda_i^a$. Our goal is then to predict the underlying individual rate parameter $\lambda_i^a$, given only bag-level observations $y^a$. To make this problem even more challenging, we embed the data manifold into $\mathbb{R}^{18}$ by rotating it with a random orthogonal matrix. For the choice of $k$ for VBAgg and Nyström, we use the RBF kernel, with the bandwidth parameter learnt. For landmark locations, we use the K-means++ algorithm, so that landmark points lie evenly across the data manifold.

**Varying number of Bags:** $n$    To see the effect of increasing number of bags available for training, we fix $N_{mean} = 150$ and $N_{std} = 50$, and vary the number of bags $n$ for the training set from 100 to 350 with the same number of bags for early stopping and validation. Each experiment is repeated for 5 runs, and results are shown in Figure 1 for individual NLL on the train set. Again we emphasise that the individual labels are not used in training. We see that all versions of VBAgg outperform all other models, in terms of MSE and NLL, with statistical significance confirmed by a signed rank permutation test (see Appendix H.1.1). We also observe that the bag-pixel model has poor performance, as a result of losing individual level covariate information in training by simply aggregating them.

**Varying number of individuals per bag:** $N_{mean}$    To study the effect of increasing bag sizes (with larger bag sizes, we expect "disaggregation" to be more difficult), we fix the number of training bags to be 600 with early stopping and validation set to be 150 bags, while varying the number of individuals per bag through $N_{mean}$ and $N_{std}$ in the negative binomial distribution. To keep the relative scales between $N_{mean}$ and $N_{std}$ the same, we take $N_{std} = N_{mean}/2$. The results are shown in Figure 1, focusing on the best performing methods in the previous experiment. Here, we observe that VBAgg models again perform better than the Nyström and NN models with statistical significance as reported in Appendix H.1.1, with performance stable as $N_{mean}$ increases.

**Discussion**    To gain more insight into the VBAgg model, we look at the calibration of our two different Bayesian models: VBAgg-Exp and VBAgg-Square. We compute their respective posterior quantiles and observe the ratio of times the true $\lambda_i^a$ lie in these quantiles. We present these in Appendix H.1.1. The calibration plots reveal an interesting nature about using the two different approximations for using $e^v$ versus $v^2$ for $\Psi(v)$. While experiments showed that the two model perform similarly in terms of NLL, the calibration of the models is very different. While the VBAgg-Square is well calibrated in general, the VBAgg-Exp suffers from poor calibration. This is not surprising, as VBAgg-Exp uses an additional lower bound on model evidence. Thus, uncertainty estimates given by VBAgg-Exp should be treated with care.

## 4.2 Malaria Incidence Prediction

We now demonstrate the proposed methodology on an important real life malaria prediction problem for an endemic country from the Malaria Atlas Project database[8]. In this problem, we would like to predict the underlying malaria incidence rate in each 1km by 1km region (referred to as a pixel), while having only observed aggregated incidences of malaria $y^a$ at much larger regional levels, which are treated as bags of pixels. These bags are non-overlapping administrative units, with $N_a$ pixels per bag ranging from 13 to 6,667, with a total of 1,044,683 pixels. In total, data is available for 957 bags[9].

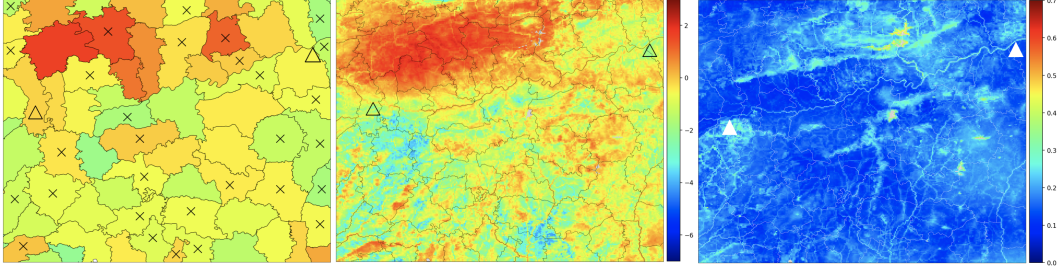

Figure 2: Triangle denotes approximate start and end of river location, crosses denotes non-train set bags. Malaria incidence rate $\lambda_i^a$ is per 1000 people. **Left, Middle**: $\log(\hat{\lambda}_i^a)$, with constant model (Left), and VBAgg-Obj-Sq (tuned on $\mathcal{L}_1^s$) (Middle). **Right**: Standard deviation of the posterior $v$ in (9) with VBAgg-Obj-Sq.

Along with these pixels, we also have population estimates $p_i^a$ (per 1000 people) for pixel $i$ in bag $a$, spatial coordinates given by $s_i^a$, as well as covariates $x_i^a \in \mathbb{R}^{18}$, collected by remote sensing. Some examples of covariates includes accessibility, distance to water, mean of land surface temperature and stable night lights. It is clear that rather than expecting malaria incidence rate to be constant throughout the entire bag (as in Figure 2), we expect pixel incidence rate to vary, depending on social, economic and environmental factors [32]. Our goal is therefore to build models that can predict malaria incidence rates at a *pixel* level.

We assume a Poisson model on each individual pixel, i.e. $y^a \sim Poisson(\sum_i p_i^a \lambda_i^a)$, where $\lambda_i^a$ is the underlying pixel incidence rate of malaria per 1000 people that we are interested in predicting. We consider the VBAgg, Nyström and NN as prediction models and use a kernel given as a sum of an ARD (automatic relevance determination) kernel on covariates and a Matérn kernel on spatial locations for the VBAgg and Nyström methods, learning all kernel parameters (the kernel expression is provided in Appendix G). We use the same kernel for manifold regularisation in the NN model. This kernel choice incorporates spatial information, while allowing feature selection amongst other covariates. For choice of landmarks, we ensure landmarks are placed evenly throughout space by using one landmark point per training bag (selected by k-means++). This is so that the uncertainty estimates we obtain are not too sensitive to the choice of landmarks. In this problem, no individual-level labels are available, so we report Bag NLL and MSE (on observed incidences) on the test bags in Appendix G over 10 different re-splits of the data. Although we can see that Nyström is the best performing method, the improvement over VBAgg models is not statistically significant. On the other hand, both VBAgg and Nyström models statistically significantly outperform NN, which also has some instability in its predictions, as discussed in Appendix G.1. However, a caution should be exercised when using the measure of performance at the bag level as a surrogate for the measure of performance at the individual level: in order to perform well at the bag level, one can simply utilise spatial coordinates and ignore other covariates, as malaria intensity appears to smoothly vary between the bags (Left of Figure 2). However, we do not expect this to be true at the individual level.

To further investigate this, we consider a particular region, and look at the predicted individual malaria incidence rate, with results found in Figure 2 and in Appendix G.1 across 3 different data splits, where the behaviours of each of these models can be observed. While Nyström and VBAgg methods both provide good bag-level performance, Nyström and VBAgg-Exp can sometimes provide overly-smooth spatial patterns, which does not seem to be the case for the VBAgg-Sq method (recall that VBAgg-Sq performed best in both prediction and calibration for the toy experiments). In particular, VBAgg-Sq consistently predicts higher intensity along rivers (a known factor [31]; indicated by triangles in Figure 2) using only coarse aggregated intensities, demonstrating that prediction of (unobserved) pixel-level intensities is possible using fine-scale environmental covariates, especially ones known to be relevant such as covariates indicated by the Topographic Wetness Index, a measure of wetness, see Appendix G.2 for more details.

In summary, by optimising the lower bound to the marginal likelihood, the proposed variational methods are able to learn useful relations between the covariates and pixel level intensities, while avoiding the issue of overfitting to spatial coordinates. Furthermore, they also give uncertainty estimates (Figure 2, right), which are essential for problems like these, where validation of predictions is difficult, but they may guide policy and planning.

# 5  Conclusion

Motivated by the vitally important problem of malaria, which is the direct cause of around 187 million clinical cases [3] and 631,000 deaths [5] each year in sub-Saharan Africa, we have proposed a general framework of *aggregated observation models* using Gaussian processes, along with scalable variational methods for inference in those models, making them applicable to large datasets. The proposed method allows learning in situations where outputs of interest are available at a much coarser level than that of the inputs, while explicitly quantifying uncertainty of predictions. The recent uptake of digital health information systems offers a wealth of new data which is abstracted to the aggregate or regional levels to preserve patient anonymity. The volume of this data, as well as the availability of much more granular covariates provided by remote sensing and other geospatially tagged data sources, allows to probabilistically disaggregate outputs of interest for finer risk stratification, e.g. assisting public health agencies to plan the delivery of disease interventions. This task demands new high-performance machine learning methods and we see those that we have developed here as an important step in this direction.

## Acknowledgement

We thank Kaspar Martens for useful discussions, and Dougal Sutherland for providing the code base in which this work was based on. HCLL is supported by the EPSRC and MRC through the OxWaSP CDT programme (EP/L016710/1). HCLL and KF are supported by JSPS KAKENHI 26280009. EC and KB are supported by OPP1152978, TL by OPP1132415 and the MAP database by OPP1106023. DS is supported in part by the ERC (FP7/617071) and by The Alan Turing Institute (EP/N510129/1). The data were provided by the Malaria Atlas Project supported by the Bill and Melinda Gates Foundation.

## Footnotes

[6]Code is available on *https://github.com/hcllaw/VBAgg*

[7]The swiss roll manifold function (for sampling) can be found on the Python *scikit-learn* package.

[8]Due to confidentiality reasons, we do not report country or plot the full map of our results.

[9]We consider 576 bags for train, 95 bags each for validation and early-stop, with 191 bags for testing, with different splits across different trials, selecting them to ensure distributions of labels are similar across sets.

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
