[Supplementary Material]

## A Aggregated Exponential Family Models

Consider an observation model of the form

$$p(y|\theta) = \exp\left(\frac{y\theta - c(\theta)}{\tau}\right)h(y,\tau), \tag{15}$$

where response $y$ is one-dimensional, $\theta$ is a natural parameter corresponding to the statistic $y$, $\tau$ is a dispersion parameter, and $h(y,\tau)$ is base measure. For simplicity, we will assume that natural parameters corresponding to the other parts of the sufficient statistic are fixed and folded into the base measure. Let $\eta$ be the corresponding mean parameter, i.e.

$$\eta = \mathbb{E}_\theta y = \int y p(y|\theta) dy$$

and $\theta = F(\eta)$ be the link function mapping from mean to the natural parameters and $G(\theta)$ its inverse. We wish to model the mean parameter $\eta = \eta(x)$ using a Gaussian process on a domain $\mathcal{X}$ together with a function $\Psi$ which transforms the GP value to the natural parameter space, i.e.

$$\eta(x) = \Psi(f(x)), \qquad f \sim \mathcal{GP}(\mu, k). \tag{16}$$

For example, the mean parameter for some models is restricted to the positive part of the real line, while the GP values cover the whole real line. We will consider the following examples:

- **Normal** (with fixed variance). $F = G = identy$ and $\Psi$ can be identity, too, as there are no restrictions on the mean parameter space.
- **Poisson**. $F(\eta) = \log \eta$, $G(\theta) = e^\theta$. $\Psi$ should take a positive value, so we consider $\Psi(v) = e^v$ or $\Psi(v) = v^2$.
- **Exponential**. $p(y|\eta) = \exp(-y/\eta)/\eta$ and $\theta = -\eta$, $F(\eta) = -1/\eta$, $G(\theta) = -1/\theta$. $\Psi$ should take a positive value, so we consider $\Psi(v) = e^v$ or $\Psi(v) = v^2$

Note that the link function $F$ is concave for all the examples above.

### A.1 Bag model

We will consider the aggregation in the mean parameter space. Namely, let $y^1, \ldots, y^n$ be $n$ independent aggregate responses for each of the $n$ bags of covariates $\mathbf{x}^a = \{x_1^a, \ldots, x_{N_a}^a\}$, $a = 1, \ldots, n$. We assume the following aggregation model:

$$y^a \sim p(y|\eta_a), \quad \eta^a = \sum_{i=1}^{N_a} w_i^a \eta_i^a = \sum_{i=1}^{N_a} w_i^a \Psi(f(x_i^a)), \quad a = 1, \ldots, n. \tag{17}$$

where $w_i^a$ are fixed weights to adjust the scales among the individuals and the bag (e.g., adjusting for population size).

We also can model individual (unobserved) variables $y_i^a$ ($i = 1, \ldots, N_a$), which follow:

$$y_i^a \sim p(y|\eta_i^a), \qquad \eta_i^a = \Psi(f(x_i^a)), \quad i = 1, \ldots, N_a, \ a = 1, \ldots, n. \tag{18}$$

Note that we consider aggregation in mean parameters of responses, not in the responses themselves. If we consider a case where underlying individual responses $y_i^a$ aggregate to $y^a$ as a weighted sum, the form of the bag likelihood and individual likelihood would be different unless we restrict attention to distribution families which are closed under both scaling and convolution. However, when aggregation occurs in the mean parameter space, the form of the bag likelihood and individual likelihood is always the same. This corresponds to the following measurement process:

- Each individual has a mean parameter $\eta_i^a$ - if it were possible to sample a response for that particular individual, we would obtain a sample $y_i^a \sim p(\cdot|\eta_i^a)$
- However, we cannot sample the individual and we can only observe a bag response. But in that case, only a single bag response is taken and depends on all individuals simultaneously. Each individual contributes in terms of an increase in a mean bag response, but this measurement process is different from the two-stage procedure by which we aggregate individual responses.

## A.2 Marginal likelihood and ELBO

Let $Y = (y^1, \ldots, y^n)$ (bag observations). With the inducing points $u = f(W)$, the marginal likelihood is

$$p(Y) = \int \int \prod_{a=1}^{n} p(y^a|\eta^a)p(f|u)p(u)dudf. \tag{19}$$

The evidence lower bound can be derived as

$$
\begin{aligned}
\log p(Y) &= \log \int \int \Big\{ \prod_{a=1}^{n} p(y^a|\eta^a)\frac{p(u)}{q(u)} \Big\} p(f|u)q(u)dudf \\
&\geq \int \int \log\Big\{ \prod_{a=1}^{n} p(y^a|\eta^a)\frac{p(u)}{q(u)} \Big\} p(f|u)q(u)dudf \\
&= \sum_{a=1}^{n} \frac{y^a}{\tau} \int F\Big( \sum_i w_i^a \Psi(f(x_i^a)) \Big)q(f)df - \int c\Big( F\Big( \sum_i w_i^a \Psi(f(x_i^a)) \Big) \Big)q(f)df \\
&\quad - \int q(u) \log \frac{q(u)}{p(u)}du, \tag{20}
\end{aligned}
$$

where $q(f) = \int p(f|u)q(u)du$.

By setting the variational distribution $q(u)$ as Gaussian, the third term is tractable. The first and second terms are however tractable only in limited cases. The cases we develop are the Poisson bag model, described in the main text, as well as the normal bag model and the exponential bag model, described below.

## A.3 Normal bag model

Here we have that $F$ is identity and $c(\theta) = \theta^2/2$, which makes both the first and the second terms tractable with the choice of $\Psi(v) = v$. However, if we consider only the aggregation in the mean parameters as above, the model essentially ignores variance on the individual level. Hence, here we opt for a different and more flexible approach, utilising the fact that normal family is closed under convolutions. Consider a bag $a$ of items $\{x_i^a\}_{i=1}^{N_a}$. Each item $x_i^a$ is assumed to have a weight $w_i^a$. At the individual level, we model the (unobserved) responses $y_i^a$ as

$$y_i^a|x_i^a \sim \mathcal{N}\left( w_i^a \mu_i^a, (w_i^a)^2 \tau_i^a \right) \tag{21}$$

where $\mu_i^a = \mu(x_i^a)$, thus $\mu_i^a$ is a *mean parameter per unit weight* corresponding to the item $x_i^a$ and it is assumed to be a function of both $x_i^a$. Similarly, $\tau_i^a$ is a variance parameter per unit weight. At the bag level, we consider the following model for the observed aggregate response $y^a$, assuming conditional independence of individual responses given covariates $\mathbf{x}^a = \{x_1^a, \ldots, x_{N_a}^a\}$:

$$y^a = \sum_{i=1}^{N_a} y_i^a, \text{ i.e. } y^a|\mathbf{x}^a \sim \mathcal{N}(w^a\mu^a, (w^a)^2\tau^a), \qquad \mu^a = \sum_{i=1}^{N_a} \frac{w_i^a}{w^a}\mu_i^a, \tau^a = \frac{\sum_{i=1}^{N_a}(w_i^a)^2\tau_i^a}{(w^a)^2} \tag{22}$$

where $\mu^a$ and $\tau^a$ are the mean and variance parameters per unit weight of the whole bag $a$ and $w^a = \sum_{i=1}^{N_a} w_i^a$ is the *total weight* of bag $a$. Note under this model formulation, the variance parameter is also aggregated unlike previously. Although we can take $\tau_i^a$ to also be a function of the covariates, here for simplicity, we take $\tau_i^a = \tau^a$ to be constant per bag (note the abuse of notation). We can now compute the negative log-likelihood (NLL) across bags (assuming conditional independence given the $\mathbf{x}^a$):

$$\ell_0 = -\log\left[\Pi_{a=1}^n p(y^a|\mathbf{x}^a)\right] = \frac{1}{2}\sum_{a=1}^{n}\left\{ \log\left( 2\pi\tau^a \sum_{i=1}^{N_a}(w_i^a)^2 \right) + \frac{\left( y^a - \sum_{i=1}^{N_a} w_i^a \mu_i^a \right)^2}{\sum_{i=1}^{N_a}(w_i^a)^2\tau^a} \right\} \tag{23}$$

where $\mu_i^a = f(x_i^a)$ is the function we are interested in, and $\tau^a$ are the variance parameters to be learnt.

We can now consider the lower bound to the marginal likelihood as below (assuming $w_i^a = 1$ here to simplify notation, while the analogous expression with non-uniform weights is straightforward):

$$\log p(y|\Theta) = \log \int \int p(y, f, u | X, W, \Theta) df\, du$$

$$= \log \int \int \left( \prod_{a=1}^{n} \frac{1}{\sqrt{2\pi N_a \tau^a}} \exp\left( -\frac{(y^a - \sum_{i=1}^{N_a} f(x_i^a))^2}{2N_a \tau^a} \right) \right) \frac{p(u|W)}{q(u)} p(f|u) q(u) df\, du$$

$$\geq \int \int \log \left\{ \prod_{a=1}^{n} \frac{1}{\sqrt{2\pi N_a \tau^a}} \exp\left( -\frac{(y^a - \sum_{i=1}^{N_a} f(x_i^a))^2}{2N_a \tau^a} \right) \frac{p(u|W)}{q(u)} \right\} p(f|u) q(u) df\, du$$

$$= -\frac{1}{2} \sum_a \int \int \left\{ \frac{(y^a)^2 - 2y^a \sum_{i=1}^{N_a} f(x_i^a) + \left( \sum_{i=1}^{N_a} f(x_i^a) \right)^2}{N_a \tau^a} \right\} p(f|u) q(u) df\, du$$

$$- \frac{1}{2} \sum_a \log(2\pi N_a \tau^a) - \int q(u) \log \frac{q(u)}{p(u|W)} du. \tag{24}$$

Using again a Gaussian distribution for $q(u)$, we have $q(f) = \int p(f|u)q(u)du$, which is a normal distribution and let $q^a(f^a)$ be its marginal normal distribution of $f^a = (f(x_1^a), \ldots, f(x_{N_a}^a))$ with mean and covariance given by $m^a$ and $S^a$ as before in (9).

Then all expectations with respect to $q(f)$ are tractable and the ELBO is simply

$$\mathcal{L}(q, \theta) = -\frac{1}{2} \sum_{a=1}^{n} \left\{ \frac{(y^a)^2 - 2y^a \mathbf{1}^\top m^a + \mathbf{1}^\top \left( S^a + m^a (m^a)^\top \right) \mathbf{1}}{N_a \tau^a} \right\} - \frac{1}{2} \sum_a \log(2\pi N_a \tau^a)$$

$$- KL(q(u)||p(u|W)). \tag{25}$$

### A.4  Exponential bag model

In this case, we have $F(\eta) = -1/\eta$. We can apply the similar argument as in Lemma 1. For any $\alpha_i > 0$ with $\sum_i \alpha_i = 1$, by the concavity of $F$,

$$\int F\left( \sum_i w_i \Psi(v_i) \right) q(v_i) dv_i = \int F\left( \sum_i \alpha_i w_i / \alpha_i \Psi(v_i) \right) q(v_i) dv_i$$

$$\geq \int \sum_i \alpha_i F\left( w_i / \alpha_i \Psi(v_i) \right) q(v_i) dv_i$$

$$= \sum_i \alpha_i \int F\left( w_i / \alpha_i \Psi(v_i) \right) q(v_i) dv_i.$$

For $F(\eta) = -1/\eta$, the last line is equal to

$$\sum_i \frac{\alpha_i^2}{w_i} \int \frac{1}{\Psi(v_i)} q(v_i) dv_i.$$

When using a normal $q$, this is tractable for several choices of $\Psi$ including $e^v$ and $v^2$. If we let $\xi_i := \int \frac{1}{\Psi(v_i)} q(v_i) dv_i$, and maximize

$$\sum_i \alpha_i^2 \frac{\xi_i}{w_i}$$

under the constraint $\sum_i \alpha_i = 1$, we obtain

$$\alpha_i = \frac{(w_i/\xi_i)}{\sum_\ell (w_i/\xi_i)}.$$

Finally, we have a lower bound

$$\int F\left(\sum_i w^i \Psi(v_i)\right) q(v_i)dv_i \geq -\frac{\sum_i(w_i/\xi_i)}{\sum_i(w_i/\xi_i)^2} \tag{26}$$

where

$$\xi_i = \int \frac{1}{\Psi(v_i)} q(v_i)dv_i.$$

which is tractable for a Gaussian variational family. Also with an explicit form of $\Psi$, it is easy to take the derivatives of the resulting lower bound with respect to the variational parameters in $q(v)$.

## B   Alternative approaches

**Constant**   For the Poisson model, we can take $\lambda_i^a = \lambda_c^a$, a constant rate across the bag, then:

$$\hat{\lambda}_c^a = \frac{y^a}{p^a}$$

hence the individual level predictive distribution is the form $y_i^a \sim Poisson(\hat{\lambda}_c^a)$, and for unseen bag $r$, $\hat{\lambda}_c^{\text{bag}} = \frac{1}{\sum_{a=1}^n p^a} \sum_{a=1}^n y^a$, with predictive distribution given by $y^r \sim Poisson(p^r \hat{\lambda}_c^{\text{bag}})$.

**bag-pixel: Bag as Individual**   Another baseline is to train a model from the weighted average of the covariates, given by $x^a = \sum_{i=1}^{N_a} \frac{p_i^a}{p^a} x_i^a$ in the Poisson case, and $x^a = \sum_{i=1}^{N_a} \frac{w_i^a}{w^a} x_i^a$ in the normal case. The purpose of this baseline is to demonstrate that modelling at the individual level is important during training. Since we now have labels and covariates at the bag level, we can consider the following model:

$$y^a|x^a \sim Poisson(p^a \lambda(x^a))$$

with $\lambda(x^a) = \Psi(f(x^a))$ for the Poisson model. For the normal model, we have:

$$y^a|x^a \sim Normal(w^a \mu(x^a), (w^a)^2\tau)$$

where $\mu(x^a) = f(x^a)$ and $\tau$ is a parameter to be learnt (assuming constant across bags). Now we observe that these models are identical to the individual model, except for a difference in indexing. Hence, after learning the function $f$ at the bag level, we can transfer the model to the individual level. Essentially here we have created fake individual level instances by aggregation of individual covariates inside a bag.

**Nyström: Bayesian MAP for Poisson regression on explicit feature maps**   Instead of the posterior based on the model (6), we can also consider an explicit feature map in order to directly construct a MAP estimator. While this method does not provide posterior uncertainty over $\lambda_i^a$, it does provide an interesting connection to the settings we have considered and also manifold-regularized neural networks, as discussed below. Let $K_{zz}$ be the covariance function defined on covariates $\{z_1, \ldots z_n\}$, and consider its low rank approximation $K_{zz} \approx \mathbf{k}_{zW} K_{WW}^{-1} \mathbf{k}_{Wz}$ with landmark points $W = \{w_\ell\}_{\ell=1}^m$ and $\mathbf{k}_{zW} = (k(z,w_1), \ldots, k(z,w_\ell))^T$. By using landmark points $W$, we have avoided computation of the full kernel matrix, reducing computational complexity. Under this setup, we have that $K_{zz} \approx \Phi_z \Phi_z^\top$, with $\Phi_z = \mathbf{k}_{zW} K_{WW}^{-\frac{1}{2}}$ being the explicit (Nyström) feature map. Using this explicit feature map $\Phi$, we have the following model:

$$f_i^a = \phi_i^a \beta, \qquad \beta \sim \mathcal{N}(0, \gamma^2 I)$$

$$y^a|\mathbf{x}^a \sim \text{Poisson}\left(\sum_{i=1}^{N_a} p_i^a \lambda(x_i^a)\right), \qquad \lambda(x_i^a) = \Psi(f_i^a),$$

where $\gamma$ is a prior parameter and $\phi_i^a$ is the corresponding $i^{th}$ row of $\Phi_{\mathbf{x}^a}$. We can then consider a MAP estimator of the model coefficients $\beta$:

$$\hat{\beta} = \text{argmax}_\beta \log[\Pi_{a=1}^n p(y^a|\beta, \mathbf{x}^a)] + \log p(\beta). \tag{27}$$

This essentially recovers the same model as in (3) with the standard $l_2$ loss regularising the complexity of the function. This model can be thought of in several different ways, for example as a weight space view of the GP ([26] for an overview), or as a MAP of the Subset of Regressors (SoR) approximation [27] of the GP when $\sigma = 1$. Additional we may include manifold regulariser as part of the prior, see discussion below about neural network.

**NN: Manifold-regularized neural networks**   The next approach we consider is a parametric model for $f$ as in [13], and search the best parameter to minimize negative log-likelihood $\ell_0$ across bags. This paper considers a neural network with parameters $\theta$ for the model $f$, and uses the back-propagation to learn $\theta$ and hence individual level model $f$. However, since we only have aggregated observations at the bag level, but lots of individual covariate information, it is useful to incorporate this information also, by enforcing smoothness on the data manifold given by the unlabelled data. To do this, following [13] and [22], we pursue a semisupervised view of the problem and include an additional manifold regularisation term [2] (rescaling with $N_{\text{total}}^2$ during implementation):

$$\ell_1 = \sum_{w=1}^{N_{\text{total}}} \sum_{u=1}^{N_{\text{total}}} (f_u - f_w)^2 k_L(x_u, x_w) = \mathrm{f}^\top \mathrm{L}\, \mathrm{f} \tag{28}$$

where we have suppressed the bag index, $N_{\text{total}}$ represents the total number of individuals, $k_L(\cdot, \cdot)$ is some user-specified kernel[10], $\mathrm{f} = [f_1, \ldots, f_{N_{\text{total}}}]^\top$, L is the Laplacian defined as $\mathrm{L} = diag(\mathrm{K_L} \mathbb{1}^\top) - \mathrm{K_L}$, where $\mathbb{1}$ is just $[1, \ldots, 1]$ and $\mathrm{K_L}$ is a kernel matrix. Although this term involves calculation of a kernel matrix across individuals, in practice we consider stochastic gradient descent (SGD) and also random Fourier features [25] or Nyström approximation (see Appendix C), with scale parameter $\lambda_1$ to control the strength of the regularisation. Similarly, one can also consider manifold regularisation at the bag level, if bag-level covariates/embeddings are available, for further details, see Appendix D.

In fact, the same regularisation can be applied to the MAP estimation with the explicit feature maps. This is equivalent to having a prior $\beta \sim \mathcal{N}(0, \sigma^2 I + (\lambda_1 \Phi^\top \mathrm{L} \Phi)^{-1})$ that is data dependent and incorporates the structure of the manifold [11].

For implementation, we consider a one hidden layer neural network, with also an output layer, for a fair comparison to the Nyström approach. For activation function, we consider the Rectified Linear Unit (ReLU).

**MAP estimation of GP**   We introduce $p(f, u) = p(f|u)p(u|W)$ and consider the posterior given by $p(u|f, y, w, \theta)$, where here the conditional distribution $f|u$ is given by:

$$f|u \sim GP(\tilde{\mu}_u, \tilde{K}), \tag{29}$$

$$\tilde{\mu}(z) = \mu_z + \mathbf{k}_{zW} K_{WW}^{-1}(u - \mu_W), \quad \tilde{K}(z, z') = k(z, z') - \mathbf{k}_{zW} K_{WW}^{-1} \mathbf{k}_{\mathbf{Wz'}}$$

where $\mathbf{k}_{zW} = (k(z, W_1), \ldots, k(z, W_\ell))^T$. Using Bayes rule, we obtain:

$$
\begin{aligned}
\log[p(u|f, y, w)] &= \log[p(y|f, u)p(f, u|X, W)] \\
&= \log[p(y|f)p(f|u, X)p(u|W)] \\
&= \sum_{a=1}^{n} y^a \log(p^a \lambda^a) + \sum_{a=1}^{n} p^a \lambda^a - \sum_{a=1}^{n} \log(y^a!) + \log(p(f|u, X)) + \log(p(u|W))
\end{aligned}
$$

where $p(f|u, X) \sim \mathcal{N}(\tilde{\mu}_u, \tilde{K})$ given by above, and $p(u|W) \sim \mathcal{N}(\mu_W, \Sigma_{WW})$, i.e.

$$\log p(f|u, X) + \log p(u|W) = -\frac{1}{2}(\log(|\tilde{K}||\Sigma_{WW}|) + (f - \tilde{\mu}_u)^\top \tilde{K}^{-1}(f - \tilde{\mu}_u) + (u - \mu_W)^\top \Sigma_{WW}^{-1}(u - \mu_W) \tag{30}$$

Here, we can not perform SGD, as the latter terms does not decompose into a sum over the data. More importantly, here we require the computation of $\tilde{K}$, which contains the kernel matrix $K$, even after the use of landmarks. This direct approach is not feasible for large number of individuals, which is true in our target application, and hence we do not pursue this method, and consider Nyström and NN as baselines.

## C   Random Fourier Features on Laplacian

Here we discuss using random Fourier features [25] to reduce computational cost in calculation of the Laplacian defined as $\mathrm{L} = diag(\mathrm{K} \mathbb{1}^\top) - \mathrm{K}$, where $\mathbb{1}$ is just $[1, \ldots, 1]$ and K. Suppose the kernel

is stationary i.e. $k_w(x - y) = k(x, y)$ (some examples include the gaussian and matern kernel), then using random Fourier features, we obtain $K \approx \Phi\Phi^\top$, where $\Phi \in \mathbb{R}^{b_N \times m}$, $b_N$ denotes the total number of individuals in the batch and $m$ denotes the number of frequencies. Now we have:

$$\mathrm{f}^\top \mathrm{Lf} \approx \mathrm{f}^\top diag(\Phi\Phi^\top \mathbb{1}^\top)\mathrm{f} - \mathrm{f}^\top \Phi\Phi^\top \mathrm{f} = \mathrm{f}^\top diag(\Phi\Phi^\top \mathbb{1}^\top)\mathrm{f} - ||\Phi^\top f||_2^2 \tag{31}$$

In both terms, we can avoid computing the kernel matrix, by carefully selecting the order of computation. Note another option is to consider Nyström approximation with landmark points $\{z_1, \dots z_m\}$, then $K \approx K_{nm}K_{mm}^{-1}K_{mn}$, where $K_{mm}$ denotes the kernel matrix on landmark points, while $K_{nm}$ is the kernel matrix between landmark and data. Then $\Phi = K_{nm}K_{mm}^{-\frac{1}{2}}$.

## D   Bag Manifold regularisation

Suppose we have bag covariates $s^a$ (note these are for the entire bag), and also some summary statistics of a bag, e.g. mean embeddings [19] given by $H^a = \frac{1}{N_a}\sum_{i=1}^{N_a} h(x_i^a)$, with some user-defined $h$. Then similarly to individual level manifold regularisation, we can consider manifold regularisation at the bag level (assuming a seperable kernel for simplicity), i.e.

$$\ell_2 = \sum_{l=1}^{n}\sum_{m=1}^{n}(F^l - F^m)^2 k_s(s^l, s^m)k_h(H^l, H^m) = \mathrm{F}^\top \mathrm{L}_{\mathrm{bag}}\mathrm{F} \tag{32}$$

where $F^a = \frac{1}{N_l}\sum_{i=1}^{N_a} f_i^a$, $k_s$ is a kernel on bag covariates $s^a$, $k_\mu$ is a kernel on $H^a$, $\mathrm{L}_{\mathrm{bag}}$ is the bag level Laplacian with the corresponding kernel, and $F = [F^1, \dots, F^n]^\top$. Combining all these terms, we have the following loss function to minimise:

$$\ell = \frac{1}{b}\ell_0 + \frac{\lambda_1}{b_N^2}\ell_1 + \frac{\lambda_2}{b_N^2}\ell_2 \tag{33}$$

where $b$ is the mini-batch size in SGD, $B_N$ is the total number of individuals in each mini-batch, $\lambda_1$ and $\lambda_2$ are parameters controlling the strength of the respective regularisation.

## E   Additional details for Poisson variational derivation

### E.1   Log-sum lemma

**Lemma 2.** *Let $v = [v_1, \dots, v_N]^\top$ be a random vector with probability density $q(v)$, and let $w_i \geq 0$, $i = 1, \dots, N$. Then, for any non-negative valued function $\Psi(v)$,*

$$\int \log\Big(\sum_{i=1}^{N} w_i\Psi(v_i)\Big)q(v)dv \geq \log\Big(\sum_{i=1}^{N} w_ie^{\xi_i}\Big),$$

*where*

$$\xi_i := \int \log\Psi(v_i)q_i(v_i)dv_i.$$

*Proof.* Let $\alpha_1, \dots, \alpha_N$ be non-negative numbers with $\sum_{i=1}^{N}\alpha_i = 1$. It follows from Jensen's inequality that

$$\int \log\Big(\sum_{i=1}^{N} w_i\Psi(v_i)\Big)q(v)dv =$$

$$\int \log\Big(\sum_{i=1}^{N}\alpha_i\frac{w_i}{\alpha_i}\Psi(v_i)\Big)q(v)dv \geq$$

$$\sum_{i=1}^{N}\alpha_i\Big[\int \log\Big(\Psi(v_i)\Big)q(v_i)dv_i + \log\frac{w_i}{\alpha_i}\Big] =$$

$$\sum_{i=1}^{N}\alpha_i\xi_i + \sum_{i=1}^{N}\alpha_i\log\frac{w_i}{\alpha_i}. \tag{34}$$

By Lagrange multiplier method, maximizing the last line with respect to $\alpha$ gives

$$\alpha_i = \frac{w_i e^{\xi_i}}{\sum_{j=1}^{N} w_j e^{\xi_j}}.$$

Plugging this to (34) completes the proof. $\qquad\square$

## E.2 A lower bound of marginal likelihood for $\Psi(f) = e^f$ and $\Psi(f) = f^2$

Using Lemma 2, we obtain that

$$\int \log\Big(\sum_{i=1}^{N} p_i^a \Psi(v_i^a)\Big) q(v^a) dv^a \geq \log\Big(\sum_{i=1}^{N} p_i^a \Psi(\xi_i^a)\Big), \tag{35}$$

where

$$\xi_i^a = \int \log \Psi(v_i^a) q_i^a(v_i^a) dv_i^a.$$

The above lower bound is tractable for the popular functions $\Psi(v) = v^2$ and $\Psi(v) = e^v$ under the normal variational distributions $q^a(v^a) \sim \mathcal{N}(m^a, S^a)$. In particular,

$$\Psi(v) = e^v: \quad \xi_i^a = \int v_i^a q_i^a(v_i^a) dv_i^a = m_i^a,$$

$$\Psi(v) = v^2: \quad \xi_i^a = \int \log(v_i^a)^2 q_i^a(v_i^a) dv_i^a = -G\left(-\frac{m_i^a}{2S_{ii}^a}\right) + \log\left(\frac{S_{ii}^a}{2}\right) - \gamma,$$

where $\gamma$ is the Euler constant and

$$G(t) = 2t \sum_{j=0}^{\infty} \frac{j!}{(2)_j (3/2)_j} t^j$$

is the partial derivative of the confluent hypergeometric function [17, 1]. However, in this work we focus on the Taylor series approximation for $\Psi(v) = v^2$, as implementation of the above bound uses a large look-up table and involves linear interpolation. Furthermore, it is suggested in experiments that the secondary lower bound proposed above in Lemma 2 can lead to poor calibration, for more details, refer to Section 4.

## E.3 KL Term

Since $q(u)$ and $p(u|W)$ are both normal distribution, the KL divergence is tractable:

$$KL(q(u)||p(u|W)) = \frac{1}{2}\Big\{Tr[K_{WW}^{-1}\Sigma_u] + \log\frac{|K_{WW}|}{|\Sigma_u|} - m + (\mu_W - \eta_u)^T K_{WW}^{-1}(\mu_W - \eta_u)\Big\} \tag{36}$$

## E.4 Taylor series approximation in the variational method

We consider the integral

$$\int \log\Big(\sum_{i=1}^{N} p_i^a (v_i^a)^2\Big) q^a(v^a) dv^a$$

where $q^a$ is $\mathcal{N}(m^a, S^a)$. We note that this can be written as $\mathbb{E}\log\|V^a\|^2$, where $V^a \sim N(\tilde{m}^a, \tilde{S}^a)$, with $P^a = diag\left(p_1^a, \ldots, p_{N_a}^a\right)$, $\tilde{m}^a = P^{a1/2}m^a$ and $\tilde{S}^a = P^{a1/2}S^a P^{a1/2}$. Note that $\|V^a\|^2$ follows a non-central chi-squared distribution. We now resort to a Taylor series approximation for

$\mathbb{E}\log\|V^a\|^2$ (similar to [29]) around $\mathbb{E}\|V^a\|^2 = \|\tilde{m}^a\|^2 + tr\tilde{S}^a$, resulting in

$$
\begin{aligned}
\mathbb{E}\log\left(\|V^a\|^2\right) &= \log\left(\mathbb{E}\|V^a\|^2\right) \\
&\quad + \mathbb{E}\left[\frac{\|V^a\|^2 - \mathbb{E}\|V^a\|^2}{\mathbb{E}\|V^a\|^2} - \frac{\left(\|V^a\|^2 - \mathbb{E}\|V^a\|^2\right)^2}{2\left(\mathbb{E}\|V^a\|^2\right)^2} + \mathcal{O}\left(\left(\|V^a\|^2 - \mathbb{E}\|V^a\|^2\right)^3\right)\right] \\
&\approx \log\left(\|\tilde{m}^a\|^2 + tr\tilde{S}^a\right) - \frac{2\tilde{m}^{a\top}\tilde{S}^a\tilde{m}^a + tr\left(\left(\tilde{S}^a\right)^2\right)}{\left(\|\tilde{m}^a\|^2 + tr\tilde{S}^a\right)^2}.
\end{aligned}
$$

As commented in [29], approximation is very accurate when $\mathbb{E}\|V^a\|^2$ is large, but the caveat is that the Taylor series converges only for $\|V\|^2 \in (0, 2\mathbb{E}\|V\|^2)$ so this approach effectively ignores the tail of the non-central chi-squared.

## F  Code

All of our models were implemented in TensorFlow, and code will be published and available for use.

## G  Additional Malaria Experimental Results

Here we provide additional experimental results for the malaria dataset. In table 1, we provide results for bag level performance for NLL and MSE with 10 different test sets (after retrial of the experiments, splitting the data across train, early-stop, validation and testing). Statistical significance was not establish for the best performing Nyström method versus the VBAgg methods, this is shown in Table 2. We further provide additional prediction/uncertainty patches for 3 different splits to highlight the general behaviour of the trained models, with further explanation and details below.

It is also noted in all cases $\lambda_i^a$ is the incidence rate per 1000 people. For VBAgg and Nyström, we use an additive kernel, between an ARD kernel and a Matern kernel:

$$
k((x, s_x), (y, s_y)) = \gamma_1 \exp\left(-\frac{1}{2}\sum_{k=1}^{18}\frac{1}{\ell_k}(x_k - y_k)^2\right) + \gamma_2\left(1 + \frac{\sqrt{3}\|s_x - s_y\|_2}{\rho}\right)\exp\left(-\frac{\sqrt{3}\|s_x - s_y\|_2}{\rho}\right)
\tag{37}
$$

where $x, y$ are covariates, and $s_x, s_y$ are their respective spatial location. Here, we learn any scale parameters and weights during training. For the NN, we also use this kernel as part of manifold regularisation, however we use an RBF kernel instead of an ARD kernel, due to parameter tuning reasons (we can no longer learn these scales).

For constant model, bag rate predictions are computed by, $p^a\hat{\lambda}_c^{\text{bag}}$, where $\hat{\lambda}_c^{\text{bag}} = \frac{1}{\sum_{a=1}^n p^a}\sum_{a=1}^n y^a$. This essentially takes into account of population.

Table 1: Results for the Poisson Model on the malaria dataset with 10 different re-splits of train, early-stopping, validation and test. Approximately, 191 bags are used for test set. Bag performance is measured on a test set, with MSE computed between $\log(y^a)$ and $\log(\sum_{i=1}^{N_a} p_i^a\hat{\lambda}_i^a)$. Brackets include standard deviation.

|  | Bag NLL | Bag MSE (Log) |
|---|---|---|
| Constant | 173.1 (31.2) | 4.08 (0.13) |
| Nyström-Exp | 88.1 (25.1) | 1.31 (0.15) |
| VBAgg-Sq-Obj | 94.1 (34.0) | 1.21 (0.05) |
| VBAgg-Exp-Obj | 97.2 (39.6) | 1.04 (0.11) |
| VBAgg-Sq | 97.6 (39.0) | 1.38 (0.18) |
| VBAgg-Exp | 99.2 (39.8) | 1.21 (0.19) |
| NN-Exp | 164.4 (127.8) | 1.82 (0.29) |

Table 2: p-values from a Wilcoxon signed-rank test for Nyström-Exp versus the methods below for Bag NLL and MSE for the malaria dataset. The null hypothesis is Nyström-Exp performs equal or worse than the considered method on the test bag performance.

|  | NLL | MSE |
| --- | --- | --- |
| Constant | 0.0009766 | 0.0009766 |
| NN-Exp | 0.00293 | 0.0009766 |
| VBAgg-Sq-Obj | 0.1162 | 0.958 |
| VBAgg-Sq | 0.1377 | 0.1611 |
| VBAgg-Exp-Obj | 0.08008 | 1.0 |
| VBAgg-Exp | 0.09668 | 0.958 |

Table 3: p-values from a Wilcoxon signed-rank test for VBAgg-Sq versus the methods below for Bag NLL and MSE for the malaria dataset. The null hypothesis is VBAgg-Sq performs equal or worse than the considered method on the test bag performance.

|  | NLL | MSE |
| --- | --- | --- |
| Constant | 0.0009766 | 0.0009766 |
| NN-Exp | 0.01855 | 0.001953 |
| VBAgg-Sq-Obj | 0.6234 | 0.9861 |
| Nyström-Exp | 0.8838 | 0.8623 |
| VBAgg-Exp-Obj | 0.6875 | 1.0 |
| VBAgg-Exp | 0.3477 | 0.9346 |

## G.1 Predicted log malaria incidence rate for various models

**Constant: Bag level observed incidences**   This is the baseline with $\hat{\lambda}_i^a$ being constant throughout the bag, as shown in Figure 3. For training, we only use $60\%$ of the data.

Figure 3: Predicted $\hat{\lambda}_i^a$ on log scale using constant model, for 3 different re-splits of the data. $\times$ denote non-train set bags.

**VBAgg-Sq-Obj**   This is the VBAgg model with $\Psi(v) = v^2$ and tuning of hyperparameters is performed based on training objective, the lower bound to the marginal likelihood, we ignore early-stop and validation set here. The uncertainty of the model seems reasonable, and we also observe that in general the areas that are not in the training set have higher uncertainties. Furthermore, in all cases, malaria incidence was predicted to be higher near the river, as discussed in Section 4.2.

Figure 4: **Top:** Predicted $\hat{\lambda}_i^a$ on log scale for VBAgg-Sq-Obj. **Bottom:** Standard deviation of the posterior $v$ in (9) with VBAgg-Sq-Obj.

**VBAgg-Sq** This is the VBAgg model with $\Psi(v) = v^2$ and tuning of hyperparameters is performed based on NLL at the bag level. Predicted incidence are similar to the VBAgg-Sq-Obj model. The uncertainty of the model is less reasonable here, this is expected behaviour, as we are tuning hyperparameters based on NLL here. In the first patch, the same parameters was chosen as VBAgg-Sq-Obj.

Figure 5: **Top:** Predicted $\hat{\lambda}_i^a$ on log scale for VBAgg-Sq. **Bottom:** Standard deviation of the posterior $v$ in (9) with VBAgg-Sq.

**VBAgg-Exp-Obj** This is the VBAgg model with $\Psi(v) = e^v$ and tuning of hyperparameters is performed based on training objective, the lower bound to the marginal likelihood, we ignore early-stop and validation set here. Predicted incidence seem to be stable in general, though some smoothness is observed. The uncertainty of the model is also not very reasonably here, but this behaviour was observed in the Toy experiments, and likely due to an additional lower bound.

Figure 6: **Top:** Predicted $\hat{\lambda}_i^a$ on log scale for VBAgg-Exp-Obj.**Bottom:** Standard deviation of the posterior $v$ in (9) with VBAgg-Exp-Obj.

**VBAgg-Exp** This is the VBAgg model with $\Psi(v) = e^v$ and tuning of hyperparameters is performed based on NLL. For details, see discussion above for the VBAgg-Exp-Obj model.

Figure 7: **Top:** Predicted $\hat{\lambda}_i^a$ on log scale for VBAgg-Exp. **Bottom:** Standard deviation of the posterior $v$ in (9) with VBAgg-Exp.

**Nyström-Exp** This is the Nyström-Exp model, it is clear that while it performs best in terms of bag NLL, sometimes prediction are too smooth in the pixel space, this is because it optimises directly bag NLL. This pattern might be seen to be unrealistic, and may cause useful covariates to be neglected.

Figure 8: Predicted $\hat{\lambda}_i^a$ on log scale for Nyström-Exp.

**NN-Exp**  We can see that the model is not very stable, this can be potentially due to the model does not have an inbuilt spatial smoothness function unlike other methods. It only uses manifold regularisation for training. Also, the maximum predicted pixel level intensity rate $\hat{\lambda}_i^a$ is over 1000 in some cases, this is clearly physically impossible given $\lambda_i^a$ is rate per 1000 people.

Figure 9: Predicted $\hat{\lambda}_i^a$ on log scale for NN-Exp.

## G.2    Remote Sensing covariates that provide the existence of a river

Here, we provide figures for some covariates that give information that there is a river as indicated by the triangles in Figure 2.

Figure 10: Topographic wetness index, measures the wetness of an area, rivers are wetter than others, as clearly highlighted.

Figure 11: Land Surface Temperature at night, river is hotter at night, due to river being able to retain heat better.

# H Additional Toy Experimental Results

In this section, we provide additional experimental results for the Normal and Poisson model. In particular, we provide results on test bag level performance, and provide also prediction, calibration and uncertainty plots.

For the VBAgg model, during the tuning process, it is possible to choose tuning parameters (e.g. learning rate, multiple-initialisations, landmark choices) based on NLL with an additional validation set or on the objective $\mathcal{L}_1$ on the training set. To compare the difference, we denote the model tuned on NLL as VBAgg and the model tuned on $\mathcal{L}_1$ as VBAgg-Obj. Intuitively, as VBAgg-Obj attempts to obtain as tight a bound to the marginal likelihood, we would expect better performance in calibration, i.e. more accurate uncertainties.

For calibration plots, we compute the $\alpha$ quantiles of the approximated posterior distribution and consider the ratio of times the underlying rate parameter $\lambda_i^a$ (or $\mu_i^a$ for the normal model) appear inside the quantiles of the posterior distribution. If the model provides good uncertainties/calibration, we should expect to see the quantiles to match with the observed ratio.

In the case of $\Psi(v) = v^2$, the approximated posterior distribution is simply a non-central $\chi^2$ distribution, while for $\Psi(v) = e^v$, this is a log-normal distribution. For the Normal Model, it is simply a normal distribution, as we do not have any transformations. Calibration plots can be found in Figure 20 and Figure 21 for the Normal Model, with Figure 14 and Figure 15 for the Poisson Model.

For uncertainty plots, we plot the standard deviation of the posterior of $v \sim \mathcal{N}(m^a, S^a)$ (i.e. before transformation through $\Psi$), as this provides better interpretability. Uncertainty plots can be found in Figure 17 and 23

To demonstrate statistical significance of our result, we aggregate the repetitions in each experiment for each method and consider a one sided rank permutation test (Wilcoxon signed-rank test) to see whether VBAgg is statistically significant better than other approaches for individual NLL and MSE.

## H.1 Poisson Model

### H.1.1 Swiss Roll Dataset

We provide additional results here for the experimental settings that we consider.

Figure 12: Varying number of bags over 5 repetitions.**Left Column:** Individual average NLL and MSE on train set. **Right Column:** Bag average NLL and MSE on test set (of size $500$). Constant prediction NLL and MSE is $2.23$ and $0.85$ respectively. bag-pixel model prediction NLL is above $2.4$ and MSE is above $3.0$, hence not shown on graph.

The varying number of bags experimental results is found in Figure 12, with the corresponding table of p-values in Table 4, 5 demonstrating statistical significance of the VBAgg-Exp and VBAgg-Sq method. Similarly, the varying number of individuals per bag through $N_{mean}$ experimental result can be found in Figure 13, with the corresponding table of p-values in Table 6, 7. The comparison between VBAgg-Exp and VBAgg-Sq was found to be non-significant.

Table 4: p-values from a Wilcoxon signed-rank test for VBAgg-Sq versus the methods below for the varying number of bags experiment for the Poisson model. The null hypothesis is VBAgg-Sq performs equal or worse than NN or Nyström in terms of individual NLL or MSE on the train set.

|  | NLL | MSE |
|---|---|---|
| NN-Exp | 6.98e−06 | 0.00025 |
| Nyström-Exp | 0.00048 | 0.00015 |

Table 5: p-values from a Wilcoxon signed-rank test for VBAgg-Exp versus the methods below for the varying number of bags experiment for the Poisson model. The null hypothesis is VBAgg-Exp performs equal or worse than NN or Nyström in terms of individual NLL or MSE on the train set.

|  | NLL | MSE |
|---|---|---|
| NN-Exp | 2.48e−06 | 2.48e−05 |
| Nyström-Exp | 0.0005 | 0.00025 |

Figure 13: Varying number of individuals per bag $N_{mean}$ over 5 repetitions. **Left Column:** Individual average NLL and MSE on train set. **Right Column:** Bag average NLL and MSE on test set (of size 500). Constant prediction NLL and MSE is 2.23 and 0.85 respectively.

Table 6: p-values from a Wilcoxon signed-rank test for VBAgg-Sq versus the methods below for the varying number of individuals per bag experiment for the Poisson model. The null hypothesis is VBAgg-Sq performs equal or worse than NN or Nyström in terms of individual NLL or MSE on the train set.

|  | NLL | MSE |
|---|---|---|
| NN-Exp | $1.81e{-}05$ | $9.53e{-}06$ |
| Nyström-Exp | 0.062 | 0.041 |

Table 7: p-values from a Wilcoxon signed-rank test for VBAgg-Exp versus the methods below for the varying number of individuals per bag experiment for the Poisson model. The null hypothesis is VBAgg-Exp performs worse than NN or Nyström in terms of individual NLL or MSE on the train set.

|  | NLL | MSE |
|---|---|---|
| NN-Exp | $6.68e{-}05$ | 0.00016 |
| Nyström-Exp | 0.049 | 0.062 |

**Calibration Plots for the Swiss Roll Dataset** In Figure 14 and 15, we provide calibration results for both experiments that we have considered. See top of Appendix H for a further details. It is clear that while VBAgg-Sq-Obj and VBAgg-Sq provides good calibration in general, this is not the case for VBAgg-Exp-Obj and VBAgg-Exp. This is not surprising as the VBAgg-Exp methods uses an additional lower bound.

Figure 14: Absolute Error in coverage from 70% to 95% for the increasing number of bags experiment for the Poisson Model. Shaded regions highlight the standard deviation. Perfect coverage would provide a straight line at 0 error.

Figure 15: Absolute Error in coverage from 70% to 95% for the increasing number of individuals per bag $N_{mean}$ and $N_{std}$ for the Poisson Model. Shaded regions highlight the standard deviation. Perfect coverage would provide a straight line at 0 error.

**Prediction and uncertainty plots**  In Figure 16 and 17, we provide some prediction plots for different models, and uncertainties for VBAgg models.

Figure 16: Individual predictions on the train set for the swiss roll dataset with 150 bags for NN and Nyström model. Here $N_{mean} = 150$, with $N_{std} = 50$.

Figure 17: Predictions and uncertainty on the swiss roll dataset with 150 bags for the VBAgg-Obj models. Here $N_{mean} = 150$, with $N_{std} = 50$. For uncertainty, we plot the standard deviation of the posterior of $v$, coming from $v^a \sim \mathcal{N}(m^a, S^a)$ in (9).

## H.2 Normal Model

### H.2.1 Swiss Roll Dataset

In this section, we provide some experimental results for the Normal model, where throughout we assume $\tau_i^a = \tau$, same for all individuals.

We consider the same swiss roll dataset as in the Poisson model, here the colour of each point to be the underlying mean $\mu_i^a$. We then consider $y_i^a \sim \mathcal{N}(\mu^a, \tau)$ with $\tau = 0.1$, hence bag observations are given by $y^a = \sum_{i=1}^{N_a} y_i^a \sim \mathcal{N}(\mu^a, N_a\tau)$ with $\mu^a = \sum_{i=1}^{N_a} \mu_i^a$. Here, the goal is to predict $\mu_i^a$ and $\tau$, given bag observations $y^a$ only. The results for the experiments are shown below in Figure 18 and Figure 19, which shows the VBAgg outperforming the NN and Nyström model. To show statistical significance, we also report the corresponding table of p-values in Table 8 and Table 9. Furthermore, we would also like to point out that the VBAgg is well calibrated as shown in Figure 20.

Figure 18: Varying number of bags over 5 repetitions for the Normal model.**Left Column:** Individual average NLL and MSE on train set. **Right Column:** Bag average NLL and MSE on test set (of size 500). Constant model individual MSE is 0.04.

Table 8: p-values from a Wilcoxon signed-rank test for VBAgg versus the methods below for the varying number of bags experiment for the Normal model. The null hypothesis is VBAgg performs equal or worse than NN or Nyström in terms of individual NLL or MSE on the train set.

|          | NLL        | MSE        |
|----------|------------|------------|
| NN       | 5.96e−07   | 4.79e−09   |
| Nyström  | 4.01e−08   | 6.52e−09   |

Table 9: p-values from a Wilcoxon signed-rank test for VBAgg versus the methods below for the varying number of individuals per bag $N_{mean}$ experiment for the Normal nodel. The null hypothesis is VBAgg performs worse than NN or Nyström in terms of individual NLL or MSE on the train set.

|          | NLL        | MSE        |
|----------|------------|------------|
| NN       | 4.77e−06   | 4.77e−06   |
| Nyström  | 4.77e−06   | 4.77e−06   |

**Calibration Plots for the Swiss Roll Dataset**    In Figure 20 and 21, we provide calibration results for both experiments that we have considered. See top of Appendix H for further details. It is clear

Figure 19: Varying number of individuals per bag $N_{mean}$ over 5 repetitions. **Left Column:** Individual average NLL and MSE on train set. **Right Column:** Bag average NLL and MSE on test set (of size $500$). Constant model individual MSE is $0.039$.

that VBAgg-Obj has better calibration in general, this is not surprising as it is tuned based on the correct objective, rather than NLL.

Figure 20: Absolute Error in coverage from $70\%$ to $95\%$ for the increasing number of bags experiment for the Normal Model. Shaded regions highlight the standard deviation. Perfect coverage would provide a straight line at $0$ error.

Figure 21: Absolute Error in coverage from $70\%$ to $95\%$ for the increasing number of individuals per bag $N_{mean}$ and $N_{std}$ for the Normal Model. Shaded regions highlight the standard deviation. Perfect coverage would provide a straight line at $0$ error.

**Prediction and uncertainty plots** Here, we provide some prediction plots for different models.

Figure 22: Individual predictions on the train set for the swiss roll dataset with 150 bags for NN and Nyström model. Here $N_{mean} = 150$, with $N_{std} = 50$.

Figure 23: Predictions and uncertainty on the swiss roll dataset with 150 bags for the VBAgg-Obj model. Here $N_{mean} = 150$, with $N_{std} = 50$. For uncertainty, we plot the standard deviation of the posterior of $v$, coming from $v^a \sim \mathcal{N}(m^a, S^a)$ in (9).

### H.2.2 Elevators Dataset

For a real dataset experiment, we consider the elevators dataset[12], which is a large scale regression dataset[13] containing 16599 instances, with each instance $\in \mathbb{R}^{17}$. This dataset is obtained from the task of controlling F16 aircraft, with the label $y$ being a particular action taken on the elevators of the aircraft $\in \mathbb{R}$. For the model formulation we assume each label follows a normal distribution, i.e. $y_l \sim \mathcal{N}(\mu_l, \tau)$, where $\tau$ is a fixed quantity to be learnt. In practice, we can imagine the action taken may differ according to the operator.

In order formulate this dataset in an aggregate data setting, we sample bag sizes from a negative binomial distribution as before, with $N_{mean} = 30$ and $N_{std} = 15$, and also take $w_i^a = 1$. To place observations into bags, similar to the swiss roll dataset, we consider a particular covariate, and place instances into bags based on the ordering of the covariate. We now have the bag-level model given by $y^a \sim \mathcal{N}(\mu^a, N_a\tau)$, with individual model $y_i^a \sim \mathcal{N}(\mu_i^a, \tau)$ and it is our goal to predict $\mu_i^a$ (and also infer $\tau$), given only $y^a$. After the bagging process, we obtain approximately 225 bags for training, and 33 bags each for early stopping, validation and testing (for bag level performance). Further, in order to neglect variables that do not provide signal, we use an ARD kernel for the VBAgg and

Table 10: Results for the Normal Model on the elevators dataset with 50 repetitions. Indiv represents individuals on train set here, while bag performance is measured on a test set. Numbers in brackets denotes p-values from a Wilcoxon signed-rank test for VBAgg versus the method. The null hypothesis is VBAgg performs equal or worse than NN or Nyström in terms of individual NLL or MSE on the train set. It is also noted MSE is computed on the observed $y_i^a$ or $y^a$, rather than the unknown $\mu_i^a$ or $\mu^a$.

| | Indiv NLL | Bag NLL | Indiv MSE | Bag MSE |
|---|---|---|---|---|
| Constant | N/A | N/A | 0.010 | 0.366 |
| VBAgg | -1.69 | 0.003 | 0.0018 | 0.052 |
| VBAgg-Obj | -1.71 | -0.02 | 0.0018 | 0.052 |
| Nyström | $-1.57(1.5e{-}13)$ | 0.003 | $0.0024\ (8.9e{-}16)$ | 0.041 |
| NN | -1.64 (0.0001258) | 0.082 | $0.0021\ (8.8e{-}10)$ | 0.041 |

Nyström model, as below:

$$k_{ard}(x, y) = \gamma_{scale} \exp\left(-\frac{1}{2}\sum_{k=1}^{d}\frac{1}{\ell_k}(x_k - y_k)^2\right) \tag{38}$$

and learn kernel parameters $\gamma_{scale}$ and $\{\ell_k\}_{k=1}^{d}$. We repeat this process and splitting of the dataset 50 times and report individual NLL results, and also MSE results in Table 10. From the results, we observe that the VBAgg model performs better the Nyström and NN model, with statistical significance.

## Footnotes

[10]In practice, this does not have to be a positive semi-definite kernel, it can be derived from any notion of similarity between observations, including k-nearest neighbours.

[11]In order to guarantee positive definiteness of Laplacian, one can add $\epsilon I$, where $\epsilon > 0$.

[12]This dataset is publicly available at http://sci2s.ugr.es/keel/dataset.php?cod=94

[13]We have removed one column that is almost completely sparse.