[Reviews · NeurIPS 2018]

Reviewer 1



In this paper, the authors propose a general framework of aggregated observation models using Gaussian process with variational methods for inference. They focus on the Poisson link function in exponential family for counts data and derive some lower bounds for optimization using variational inference, Taylor expansion and Jensen's inequality. They apply the methods on synthetic data and Malaria Atlas data to show the performance. In this work, the authors try to predict the labels for individuals while only aggregated labels are available. This problem is studied in Ref. [13, 14, 20]. In this work, the authors extend previous works to exponential family and bring in Gaussian process which can derive uncertainty. Overall, I think this is a good paper. The theorems and experiments are comprehensive and solid. The extension to use Poisson distribution can be useful for count data and the usage of Gaussian process enables one to get the uncertainty. The paper is well-written. Below are some suggestions which I think can make the paper more clear. 1. For equation (7) in line 145, please define what is z. 2. For equation (6) in line 144, please reformulate the equation. Currently, there exists \prod_{i=1}^{N_a} but i does not appear in the equation. 3. In variational inference, the authors choose the variational posterior $q(f) = \int p(f|u)q(u)du$, which shares the conditional distribution $p(f|u)$ with the prior definition. Though the authors have presented that in line 156, I prefer the authors to discuss their choice of variational posterior $q(f)$ before presenting the lower bound in line 149. 4. In Section 4. Experiments, it seems that the authors compare their methods VBAgg with two other approaches Nystrom and NN. Could the authors provide reference for Nystrom and NN and explain more about the formulation of these two approaches? In practice, how can one know the 'population' $p^a_i$ in line 113? Could the authors explain more about how they get $p^a_i$ in line 264? Could $p^a_i$ be treated as parameters to learn as well? Note: I have read the feedback. Thanks.

Reviewer 2



PAPER SUMMARY: This paper proposes an interesting variational inference framework for a weakly supervised learning scenario (with counting data) where the corresponding output to each input datum is not directly observable. Instead, the training data is partitioned into subsets & a noisy observation of the aggregate output of each subset is provided. The task is to learn a mapping from each (unseen) input datum to its corresponding output. SIGNIFICANCE & NOVELTY: Addressing the regression task using aggregate observations instead of individual observations using GP with Poisson likelihood is interesting & fairly new to me. The technical development also appears correct and feasible. The authors should, however, provide a more in-depth discussion to highlight the fundamental differences (in terms of both model formulation & solution technique) between their proposed framework & the previous work of Gaussian process modulated Poisson process [17] (VBPP). This is only mentioned very briefly in the related work section that VBPP did not handle aggregation of output. While this is true, it is also interesting to know why it is non-trivial to extend VBPP to handle aggregate output & how did the authors' new formulation & solution technique address(or sidestep) this challenge (hence, the significance of the proposed work). EXPERIMENT: A related work [17] (VBPP) was also tested on the same domain of Malaria prediction task. Is the experiment setting in [17] amenable to the proposed framework (e.g., one datum per bag)? It will be interesting to compare the performance of VBAgg and VBPP in such setting. The authors may also want to include further experiments with varying the no. of inducing points. As we increase the no. of inducing points, the prediction quality should improve at the cost of extra processing time. For a specific task, I imagine this would help us determining the "right" no. of inducing points (i.e., too few inducing points will degrade the prediction quality but too many inducing point will incur extra computing cost). CLARITY: The paper is clearly written. I do, however, find the notations are a bit too dense & hard to follow. If possible, please consider simplifying the notations to improve readability. REVIEW SUMMARY: This is an interesting paper with novel contributions. The proposed theoretical framework is new to me (to the best of my knowledge) & is also demonstrated empirically in an important real-life problem with promising results. ---- I have read the rebuttal. Thank you for the clarifications.

Reviewer 3



Summary: this submission considers Gaussian process regression when only aggregate outputs are available, that is individual outputs are not reported but grouped or bagged together. The paper proposes a likelihood model based on exponential families, assuming the mean parameters for the aggregate likelihood is the weighted sum of those of individual likelihoods. Inference and learning are made tractable using the popular sparse variational approximations based on pseudo-points. Additional intractable expectations in the lower bound are sidestepped using Taylor expansion or additional lower bounding. Some experiments were provided to support the proposed model and inference scheme. Comments: The paper only focused on the Poisson case and developed additional approximations specific to this case. However, it is unclear how to move beyond these models (Poisson, exponential or normal [with very specific constraints]), say classification with binary aggregate outputs?. Only the Poisson case was demonstrated on real data. The Titsias's variational approximation based on inducing points used here is, perhaps, well-known, and the approximation presented in this paper is a straightforward extension of Titsias's method for aggregate likelihoods. The Poisson model was tested on one toy and one real world dataset. On the toy dataset, the results seem to be mixed and that the Nystrom method is doing quite well. On the real dataset, it is hard to judge quantitatively how well each model performs as the numbers in the appendix are really close [not to mention Bayesian linear regression with Nystrom features is again doing well here] and the standard errors are quite big, though I agree that, looking at the figures, the predictions using the VBAgg-Sq might be more interpretable. It might be interesting and informative if this was tested on another dataset where testing can be done at the individual pixel level, as well as the bag level. Clarity: The paper is generally well written. The notations in section 3.1 are quite heavy. Note: I have read the author response and will keep my original rating and evaluation as is. The work in this submission is solid, but would be great if the proposed model + inf method could be evaluated on some extra real datasets.